# Effects of 5-Week of FIFA 11+ Warm-Up Program on Explosive Strength, Speed, and Perception of Physical Exertion in Elite Female Futsal Athletes

**DOI:** 10.3390/sports10070100

**Published:** 2022-06-27

**Authors:** Antonino Patti, Valerio Giustino, Stefania Cataldi, Vito Stoppa, Federica Ferrando, Riccardo Marvulli, Giacomo Farì, Şahin Fatma Neşe, Antonino Bianco, Antonella Muscella, Gianpiero Greco, Francesco Fischetti

**Affiliations:** 1Sport and Exercise Sciences Research Unit, Department of Psychology, Educational Science and Human Movement, University of Palermo, 90144 Palermo, Italy; antonino.patti01@unipa.it (A.P.); valerio.giustino@unipa.it (V.G.); antonino.bianco@unipa.it (A.B.); 2Department of Basic Medical Sciences, Neuroscience and Sense Organs, University of Study of Bari, 70124 Bari, Italy; federica.ferrando@hotmail.com (F.F.); riccardo.marvulli@policlinico.ba.it (R.M.); gianpiero.greco@uniba.it (G.G.); francesco.fischetti@uniba.it (F.F.); 3FIGC Italian Football Federation, Puglia Regional Committee—Futsal Division, 70124 Bari, Italy; vitostoppa@msn.com; 4Department of Biological and Environmental Science and Technologies, University of Salento, 73100 Lecce, Italy; giacomo.fari@unisalento.it (G.F.); antonella.muscella@unisalento.it (A.M.); 5Department of Sport and Health, Faculty of Sport Sciences, Ankara University, 06830 Ankara, Turkey; nesesahin@ankara.edu.tr

**Keywords:** sport performance, soccer, FIFA 11+, warm-up, Squat Jump, vertical jump height, jumping performance, Agility T-test

## Abstract

Futsal is a sport that originates from soccer and is increasingly practiced all over the world. Since training and warm-up protocols should be sport-specific in order to reduce injuries and maximize performance, this study aimed to evaluate the effects of 5 weeks of the FIFA 11+ warm-up program on explosive strength, speed, and perception of physical exertion in elite female futsal athletes. Twenty-nine elite female futsal athletes participating in the Italian national championships were divided into two groups: the experimental group (EG) underwent 5 weeks of the FIFA 11+ warm-up program, and the control group (CG) underwent 5 weeks of a dynamic warm-up. We evaluated any effect on explosive strength (by Squat Jump test), speed (by Agility T-test), and perception of physical exertion (by Borg CR-10 scale). All measurements were carried out by a technician of the Italian Football Federation before (T0), at the middle (T1), and at the end (T2) of the protocol. The EG showed significant improvements on performances between T0 vs. T1 and T0 vs. T2 both in the Squat Jump test (*p* = 0.0057 and *p* = 0.0030, respectively) and in the Agility T-test (*p* = 0.0075 and *p* = 0.0122). No significant differences were found in the Squat Jump test performances in the CG, while significant improvements were detected in the Agility T-test performances (*p* = 0.0004 and *p* = 0.0053, T0 vs. T1 and T0 vs. T2, respectively). As for the Borg CR-10 scale, we found a significant difference between T0 and T2 in the EG (*p* = 0.017) and no differences in the CG. This study showed that 5 weeks of the FIFA 11+ warm-up program improves the jumping performance of female futsal athletes without adversely affecting speed. These findings can be useful for coaches and athletic trainers in order to consider FIFA 11+ warm-up program also in female futsal athletes.

## 1. Introduction

Warm-up is considered an essential factor for preventing muscle injury and for improving performance in athletes [1,2]. The Fédération Internationale de Football Association 11+ (FIFA 11+) is a warm-up program that is composed of three parts that include 15 exercises, and its application has been primarily focused on preventing and reducing injuries in soccer players [3,4]. Data have also indicated improvements on physical performance following the implementation of the FIFA 11+ program [5]. As a matter of fact, Zarei et al. (2018) showed higher improvements on agility, vertical jump, and speed performances in favor of the FIFA 11+ program compared to a traditional warm-up [6]. Chen et al. (2019) observed that, although neuromuscular performance slightly decreased immediately after FIFA 11+ warm-up program, it slightly increased 10, 20, and 30 min after the end [7]. Another study demonstrated significant improvements on agility and long-jump performances after 4 weeks of the FIFA 11+ warm-up program also in young soccer players [8]. However, few studies have evaluated the effects of FIFA 11+ in futsal players [8,9]. Lopes et al. (2019) revealed no improvements in static, dynamic balance, and proprioception in amateur futsal players after 10 weeks of the FIFA 11+ program [9]. In contrast, in 2020, the same research group reported significant long-term benefits on eccentric strength by applying the FIFA 11+ program for 10 weeks [10]. Even fewer studies have investigated the effects of the FIFA 11+ warm-up program on performance in female players, especially in futsal [11,12]. In this sport, as for other team sports, strength and jumping performances are essential for competition [13,14,15].

As demonstrated by the different types of warm-ups in the literature, with sometimes conflicting results, warm-up is a common practice among athletes. For instance, many coaches recommend their athletes to perform stretching before exercise. This recommendation is based on the idea that stretching improves performance, prevents injuries, and increases flexibility. Thomas et al. (2018) examined different strategies and protocols of stretching and found that all types of stretching showed long-term Range of Motion improvements; however, static protocols showed significant gains compared to the ballistic or Proprioceptive Neuromuscular Facilitation protocols [16]. In contrast, some authors have reported that static stretching before exercise can lead to decreased performance [17]. A previous research study indicated that a warm-up that includes static stretching may adversely affect jumping performance but not sprint time [17]. In another study conducted to compare the effects of static and dynamic stretching on vertical jump performance and vastus medial electromyographic activity in males who participated in competitive university sports, the authors found that static stretching has a negative influence on vertical jump performance, while dynamic stretching has a positive impact on it, with a significantly greater electromyographic amplitude in the dynamic stretching compared with the static stretching [18]. McMillian et al. (2006) compared the effects of static stretching and dynamic stretching on power and agility performance, detecting better scores in the participants who had performed dynamic stretching [19].

As for futsal, Tomsovsky et al. (2021) examined the effectiveness of a specific warm-up consisting of cardiovascular exercises, dynamic stretching, and game-related skills to reduce injuries in amateur players, finding a reduction in the rate of all injuries [20].

Based on these premises, the literature reports many studies that have analyzed different types of and protocols for warm-up, showing conflicting results [21,22]. However, there is little information regarding the effects of warm-up in female futsal players. Therefore, this study aimed to evaluate the effects of 5 weeks of the FIFA 11+ warm-up program on explosive strength, speed, and perception of physical exertion in elite female futsal athletes, comparing it with a dynamic warm-up.

## 2. Materials and Methods

### 2.1. Study Design

This is a non-randomized controlled study in which the experimental group (EG) was given 5 weeks of the FIFA 11+ warm-up program, while the control group (CG) was given 5 weeks of a dynamic warm-up in order to evaluate any effect on explosive strength (measured through the Squat Jump test), speed (measured through the Agility T-test), and perception of physical exertion (measured through the Borg CR-10 scale).

All measurements were carried out by a technician of the Italian Football Federation before (T0), at the middle (T1), and at the end (T2) of the protocol. All tests, carried out in the futsal field used for training, were administered in the same order. All measurements were carried out at the same time of the day in order for each participant to avoid any circadian-rhythm-related variations, and 48 h after the last training in order to minimize the influence of fatigue. During tests administration, all participants were dressed in shorts, shirts, and athletic shoes.

Participants were recruited by two teams from the Puglia Region in Italy (City of Taranto, ASD Dona Five Fasano) who took part in the Italian national championships.

A written informed consent was obtained from each participant before her participation in the study. The study was carried out in compliance with the principles of the Declaration of Helsinki and approved by the Independent Ethics Committee of the Bari University Hospital (code 7238, 11 March 2022).

### 2.2. Participants

Twenty-nine elite female futsal athletes participated voluntarily in this study. An a priori sample size power analysis with an α error of 0.05 and an effect size of 0.25 revealed that 28 participants in total would be sufficient to reach a power of 80%.

Participants were divided into two groups: the experimental group (EG) underwent 5 weeks of the FIFA 11+ warm-up program, and the control group (CG) underwent 5 weeks of the dynamic warm-up. The EG was composed of 14 athletes (age: 26.21 ± 7.38 years; height: 160.07 ± 4.30 cm; weight: 60.81 ± 6.26 kg); the CG was composed of 15 athletes (age: 26.80 ± 6.52 years; height: 165.07 ± 7.61 cm; weight: 63.31 ± 5.30 kg). Weight was measured by using a scale to the nearest 100 g (Wunder 960 classic; Trezzo sull’Adda, Milan, Italy). Height was measured by using a portable stadiometer that is sensitive to changes of 1 cm (Seca 220; Hamburg, Germany). Each measurement was performed twice, and the arithmetic mean was recorded [23].

The study envisaged the following exclusion criteria: male athletes and athletes who had suffered lower- and/or upper-limb injuries in the past 2 years.

### 2.3. Protocol

#### 2.3.1. FIFA 11+ Warm-Up Program

The EG underwent 5 weeks of FIFA 11+ warm-up program, 2 sessions/week, performed in the specific sequence recommended by the manual. The program consisted of the following three parts for a total of 15 exercises: (1) running exercises (8 min); (2) strength, plyometrics, and balance exercises (10 min); and (3) running exercises (2 min).

#### 2.3.2. Dynamic Warm-Up

The CG underwent 5 weeks of dynamic warm-up, 2 sessions/week. The program consisted of the following four parts: (1) running exercises (5 min); (2) upper limbs’ mobility exercises (5 min); (3) lower limbs’ mobility exercises (5 min); and (4) dynamic routine, high-knee, skip, butt-kick, and grapevine exercises (5 min).

### 2.4. Measures

#### 2.4.1. Squat Jump Test

The Squat Jump test was administered on a resistive–capacitive platform (Ergojump, Psion XP, MA.GI.CA., Rome, Italy) and connected to a digital timer (accuracy: ±0.001 s) that is capable of recording the flight time and contact time of each jump. The jump height (cm) was calculated from the flight time. Each participant performed three jump trials, separated by 1-min intervals, and the highest jump height was used for statistical analysis [24].

#### 2.4.2. Agility T-Test

The Agility T-test was administered by using the modified Semenick procedure (Figure 1) [25]. Each participant performed the test, starting from point A and sprinting forward to point B (9.14 m). Subsequently, each participant performed a side run to the left toward point C (4.57 m) and then in the opposite way to point D (9.14 m). Finally, each participant, by running backward, reached point A. Three trials were performed, and the best was used for the statistical analysis [26].

#### 2.4.3. Borg CR-10 Scale

The Borg CR-10 scale was administered to evaluate the perception of physical effort at the end of the warm-up. The evaluations were carried out two times: at T0 and at T2. Participants were given detailed instructions on how to evaluate the experience regarding the perception of physical effort. Each participant evaluated the perception of physical effort on a scale from “absolutely nothing” (value 0) to “extremely strong” (value 10) [27].

### 2.5. Statistical Analysis

All data were recorded in an Excel file. Shapiro–Wilk normality test was used to analyze data distribution. Repeated-measures analysis of variance and Dunnett’s multiple comparisons test were used for comparisons. Fixed effects (type III) were used to evaluate the effect of the warm-up between the outcome measures (Squat Jump test and Agility T-test). The Wilcoxon signed-rank test was used to detect differences between the perceptions of physical effort (Borg CR-10 scale).

The statistical analysis was performed by using Statistica software version 8.0 (StatSoft Inc., Tulsa, OK, USA) and GraphPad Prism software version 5.0 (GraphPad Software, San Diego, CA, USA). Statistical significance was set a priori at *p* < 0.05.

## 3. Results

Shapiro–Wilk normality test showed a non-Gaussian distribution of the Borg CR-10 scale (*p* < 0.05). Conversely, the Squat Jump test and the Agility T-test showed a Gaussian distribution (*p* > 0.05). Table 1 shows the analysis of explosive strength and speed variables before (T0), at the middle (T1), and at the end (T2) of the warm-up protocols for both groups. Dunnett’s multiple comparisons analysis showed that, both in the Squat Jump test and in the Agility T-test EG showed significant improvements in performance between T0 vs. T1 and T0 vs. T2. The fixed effects analysis showed a significant statistical difference for the Squat Jump test and the Agility T-test (*p* < 0.05). In the CG, no significant differences were found in the Squat Jump performances, while significant improvements were detected in the Agility T-test performances (*p* < 0.05).

As for the Borg CR-10 scale, the Wilcoxon signed-rank test showed a significant difference between T0 and T2 in the EG (*p* < 0.05), as reported in Table 2. No differences were detected in the CG (*p* > 0.05).

## 4. Discussion

This study aimed to evaluate the effects of 5 weeks of the FIFA 11+ warm-up program on explosive strength, speed, and perception of physical exertion in elite female futsal athletes, comparing it with a dynamic warm-up.

Our results showed that the FIFA 11+ warm-up program led to significant improvements in performances between T0 vs. T1 and T0 vs. T2 both in the Squat Jump test and in the Agility T-test. In contrast, after a dynamic warm-up, we detected no significant differences in the Squat Jump performances, while significant improvements were found in the Agility T-test performances.

Futsal is a dynamic sport that is characterized by speed in changes of direction. In this sport, it is necessary to combine intermittent high-intensity actions and decision-making processes [28]. The warm-up plays an important role in meeting these demands because it affects players’ performance in the acute and the long-term period [29,30].

Our findings suggest that both the FIFA 11+ warm-up program and the dynamic warm-up seem to have a positive effect on agility performances, but the same trend does not show up in jumping performances. A possible explanation for the present results could be that the FIFA 11+ warm-up program included balance exercises and core-strength exercises. In fact, the vertical jump, while representing an indicator of explosive strength [31,32], is influenced by the ability to balance in its various expressions [31]. Additionally, hamstring exercises and squats improve the concentric and eccentric muscle strength of the lower limbs. Moreover, Liu et al. (2021) showed that the FIFA 11+ warm-up program can directly contribute to improving the performance of the change of direction [33].

As for the Borg CR-10 scale, we found a significantly higher perception of physical exertion between T0 and T2 when administering the FIFA 11+ warm-up program and no differences when administering the dynamic warm-up, thus indicating this component should be considered in the administration of a warm-up protocol. We speculate that this could be due to the higher intensity of the FIFA 11+ warm-up program, which also includes strength and plyometrics exercises. Faelli et al. investigated the effects of three different warm-ups including different stretching protocols such as static, dynamic, and no-stretching on several outcomes, including the perception of effort [34]. The authors found a significantly lower perception of effort in the warm-up including static and dynamic stretching compared to no-stretching.

This study showed that FIFA 11+ warm-up program improves jumping performance in female futsal players without adversely affecting speed. Hence, our findings indicated that FIFA 11+ warm-up program does not result in any decrease in performance in terms of explosive strength for the lower limbs. Silva et al., demonstrated that there should be a balance between closed- and open-skills exercises, paying attention to specific open-skills exercises, which generate optimal pre-match contexts, but, at the same time, inserting specific closed-skills exercises [30].

This study has some strengths, as it adds new knowledge on the application of FIFA 11+ warm-up program in futsal and, in particular, in female players, considering the few articles in the literature. However, this study has limitations that need to be mentioned. First, the study design is not randomized, thus limiting its internal validity. Second, although the a priori sample size power analysis determined twenty-eight participants, it would be necessary to increase the sample size in future studies.

## 5. Conclusions

In summary, the present study showed that the FIFA 11+ warm-up program improves jumping and speed performances. These findings can be useful for coaches and athletic trainers in order to consider FIFA 11+ warm-up program also in female futsal athletes.

## Figures and Tables

**Figure 1 sports-10-00100-f001:**
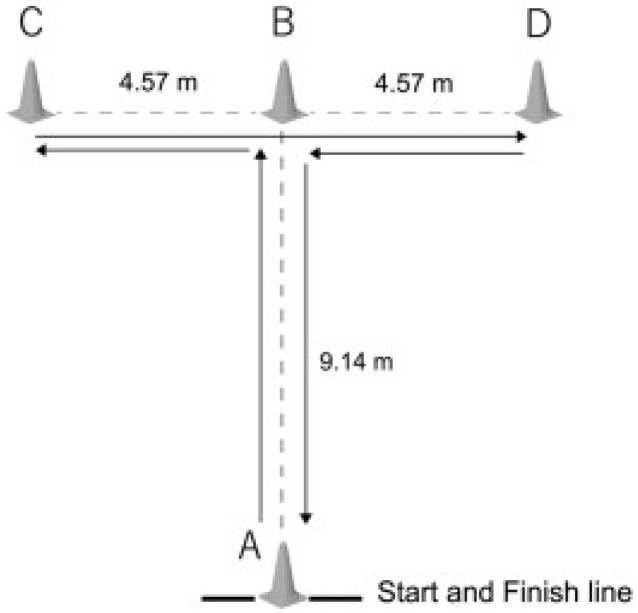
The Agility T-test. A, start and arrival point; A–B, sprinting forward; B–C, side run to the left; C–D, side run to the right; B–A, running backward.

**Table 1 sports-10-00100-t001:** Analysis of explosive strength and speed variables before (T0), at the middle (T1), and at the end (T2) of the warm-up protocols for both groups.

**EG (*n* = 14)**	**T0**		**T1**		**T2**		**Dunnett’s Multiple Comparison**		**Fixed Effect (*p* < 0.05)** **Treatment**
**Test**	**Mean**	**SD**	**Mean**	**SD**	**Mean**	**SD**	**T0 vs. T1**	**T0 vs. T2**	
Squat Jump test (cm)	32.71	6.36	33.93	5.59	35.86	5.93	0.0057 **	0.0030 **	0.0024 **
Agility T-test (s)	8.76	0.32	8.62	0.34	8.44	0.27	0.0075 **	0.0122 *	0.0077 **
**CG (*n* = 15)**	**T0**		**T1**		**T2**		**Dunnett’s multiple comparison**		**Fixed effect (*p* < 0.05)** **Treatment**
**Test**	**Mean**	**SD**	**Mean**	**SD**	**Mean**	**SD**	**T0 vs. T1**	**T0 vs. T2**	
Squat Jump test (cm)	23.40	1.68	23.60	1.88	23.67	1.63	0.8318	0.5186	0.6087
Agility T-test (s)	11.59	1.34	10.77	1.37	10.93	1.50	0.0004 ***	0.0053 **	0.0002 ***

Legend: EG, experimental group; CG, control group; T0, pre-test; T1, at the middle; T2, post-test; SD, standard deviation; * *p* < 0.05; ** *p* < 0.01; *** *p* < 0.001.

**Table 2 sports-10-00100-t002:** Analysis of perception of physical exertion variable before (T0), at the middle (T1), and at the end (T2) of the warm-up protocols for both groups.

	T0		T2		Wilcoxon Test
	Mean	SD	Mean	SD	
Borg CR-10 scale (EG)	3.35	1.44	4.5	1.45	0.017 *
Borg CR-10 scale (CG)	7.93	0.70	8	0.65	ns

Legend: EG, experimental group; CG, control group; T0, pre-test; T1, at the middle; T2, post-test; SD, standard deviation; * *p* < 0.05

## Data Availability

The data presented in this study are available upon request from the last author.

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
