# Peer review of "Effects of 5-Week of FIFA 11+ Warm-Up Program on Explosive Strength, Speed, and Perception of Physical Exertion in Elite Female Futsal Athletes"

_sports, 2022, doi:10.3390/sports10070100_

Round 1

Reviewer 1 Report

Manuscript Review

Effects of 5-week of FIFA 11+ Warm-up on Explosive Strength, 2 Speed, and Perception of Physical Exertion in Elite Female Futsal Athletes

First of all, thank you for the opportunity to review your work. Second, I would like to commend all the authors for undertaking studies in female athletes, and need to continue to push this area of research. Well-done on the manuscript, I feel this can be improved and perhaps in future, I would recommend you consider having an English native on your team to review any translation issues at the end, I think with the Borg Scale usage may have been a little issue and the wordiness within your manuscript could have been improved prior to submission.

Abstract:

 I think it is important to report which direction the differences were observed as well as some statistical results i.e. p values? Effect sizes, etc…

Please include further detail about direction, statistical difference, and perhaps descriptive data.

I am slightly confused but, do these results show that FIFA 11+ improve jumping performance or not. The “may” in this sentence makes it seem inconclusive.

Introduction

L46 – avoid having Furthermore, the literature – I suggest simply saying “Data” suggests or Data indicates … the furthermore is a little superfluous. Again, you have “also” in the sentence. Consider revising sentence altogether

L50 – you have another furthermore, - consider revising

L50-51 – if you re-read this sentence again, you will notice how wordy it is. Starting a sentence with “Particularly interesting is the study BY Chen et al. (2019) in which the authors showed that immediately…..   Can you see how you don’t need a lot of the words there – see example

Chen et al. (2019) observed that …..

Basically you say the same thing but it is a lot more concise.

L53 – you have In addition, another –

Can you see how these both mean the same thing? You can simply say Another study ….

Or a suggestion might be to combine the references and state that Both studies improved performance outcomes following the implementation of the FIFA 11.

L56 – When you say aa sentence like Few studies …. You need to ensure you are referencing the few studies at the end of the sentence.

L56 – suggestion – Lopes et al. (2019) revealed no improvements in static, balance and proprioception in amateur futsal players after a 10 week FIFA 11 + program. In contrast, …..

Or alternatively, you can simply say that the data is controversial with mixed results.  

L64 – avoid In 2018, a review by Thomas et al. – very wordy – cut straight to it.

Thomas et al. (2018) examined….

L68 – avoid starting new paragraphs with “On the other side” makes the readers think on the other side of what exactly?

L76 – again, avoid having the year then the author in the sentence – for the reminder of the paper I won’t comment on these again, but, I think you should amend all of the sentences like that. You do this throughout – please amend.

L81 – “as shown” – avoid starting new paragraph with As shown ….

At present, the data examining warm-ups has produced conflicting results, with little information regarding mixed-warm up protocol …. Therefore, the aim of this study was to evaluate the effects of a 5-week FIFA 11+ program on ……

This is just a suggestion as to how you might tighten your work.

Methods

The Borg scale reference seems incorrect – The reference used was for office exercise training and the paper reference [23] – does not reference the anchors and if it is a Borg Scale of RPE then should reference the original paper “Psychophysical bases of perceived exertion” –

The anchor you have in this paper are “absolutely nothing” to “extremely strong” which does not seem representative of RPE. Imagine asking an athlete how hard they worked and they respond “extremely strong” doesn’t quite make sense. I get the feeling that this is a translation problem, sometimes from Italian to English the actual word for word translation does not translate well. This is a normal problem.

Results

L175  - important to include the direction, higher, lower, decrease, increase  

Tables are presented well – perhaps in the methods section you need to include what is better or worse performance in the methods sections.  

Combine first and second paragraph as the first paragraph is only 2 sentences

Perhaps include a figure to represent and show the differences instead of just tables. Draw the attention of your work with a figure would be better.

Discussion

A similar theme here where you are stating the differences but, no indication as to which way, better or worse. This is also your opportunity to state which is best for others using your work or trying to replicate it.

L199 – Consider revising first sentence – you have dynamic sport then dynamism – means the same thing.

L201/2 state why the warm up is important –

One question that comes to mind here is while the static stretching did not negatively impact performance – was there anything report on the injuries obtained – I understand the paper is evaluating the warm up on performance but, is a warm-up actually designed to prepare the body to perform as well as reduce the incidence or chance of injury. I think this is something that at least needs to be discussed.

L22 – consider revising “Probably” – This is largely speculative and wordy thereafter

L224 – type Silva et al – needs a .

L228 -revise sentence is a little strange and wordy. Perhaps not even need. Simply go to the acknowledgement of limitations

L233 – why ?

I feel the limitations may not necessarily apply e.g. 3rd limitation isn’t really much – perhaps say something about the level of athlete instead

Conclusions

Final sentence can be improved.

Reviewer 2 Report

Dear authors,

I reviewed the research entitled ‘Effects of 5-week of FIFA 11+ Warm-up on Explosive Strength, Speed, and Perception of Physical Exertion in Elite Female Futsal Athletes’. I have some comments as below. Please consider them for your decision.

1) I believe that ‘warm-up' or ‘FIFA 11+' need to be included as one of the key word.

2) In line 94, when was Borg CR10 measured ? Please explain these average period. What was the purpose of this measurement? I could not find out the relationship between effect of warm-up and subjective exhaustion. 

3) In line 106, It might be error in writing that type 2 error rate is 0.8 in the method.

4) How often each intervention per week? 

5) In Table 2, experimental group (FIFA11+) of Borg CR10 increased at T2 compared to T0 significantly. It means FIFA11+ has a possibility to exhaust  players. I couldn`t find out your discussion in this point. Please explain it.

6) Although you referred to effects of static stretching in the discussion in line 216. However, I could not find out which phase was applicable as static stretching intervention.

7) Totally, I want to make sure that the purpose of  this research was to clarify whether some performances were improved by warm-up.  However, warm-up is not for training but for conditioning. I believe these outcomes are not suitable for evaluating effects of warm-up. Effects of warm-up will keep onset of injuries down during activities won't it?

8) As a whole, the tables and figures seem to be insufficient in legend.

That's all.

Round 2

Reviewer 2 Report

Dear authors,

I reviewed the research entitled 'Effects of 5-week of FIFA 11+ Warm-up Program on Explosive 2 Strength, Speed, and Perception of Physical Exertion in Elite 3 Female Futsal Athletes'.

I have no further comment. I understand all of authors reply.

That's all.

Author Response

Dear reviewer,

thank you very much for your help!